# Maternal Biomarkers of Acetaminophen Use and Offspring Attention Deficit Hyperactivity Disorder

**DOI:** 10.3390/brainsci8070127

**Published:** 2018-07-03

**Authors:** Yuelong Ji, Anne W. Riley, Li-Ching Lee, Xiumei Hong, Guoying Wang, Hui-Ju Tsai, Noel T. Mueller, Colleen Pearson, Jessica Thermitus, Anita Panjwani, Hongkai Ji, Tami R. Bartell, Irina Burd, M. Daniele Fallin, Xiaobin Wang

**Affiliations:** 1Center on the Early Life Origins of Disease, Department of Population, Family and Reproductive Health, Johns Hopkins University Bloomberg School of Public Health, 615 N Wolfe St, Baltimore, MD 21205, USA; yji7@jhu.edu (Y.J.); ariley1@jhu.edu (A.R.); xhong3@jhu.edu (X.H.); gwang24@jhu.edu (G.W.); 2Department of Epidemiology, Johns Hopkins University Bloomberg School of Public Health, 615 N Wolfe St, Baltimore, MD 21205, USA; llee38@jhu.edu (L.-C.L.); noeltmueller@jhu.edu (N.T.M.); dfallin@jhu.edu (M.D.F.); 3Wendy Klag Center for Autism and Developmental Disabilities & Department of Mental Health, 615 N Wolfe St, Baltimore, MD 21205, USA; 4Division of Biostatistics and Bioinformatics, Institute of Population Health Sciences, National Health Research Institutes, Zhunan 35053, Taiwan; tsaihj@nhri.org.tw; 5Department of Pediatrics, Boston University School of Medicine and Boston Medical Center, 1 Boston Medical Center Pl, Boston, MA 02118, USA; Colleen.Pearson@bmc.org (C.P.); Jessica.Thermitus@bmc.org (J.T.); 6Department of International Health, Johns Hopkins University Bloomberg School of Public Health, 615 N Wolfe St, Baltimore, MD 21205, USA; apanjwani@jhu.edu; 7Department of Biostatistics, Johns Hopkins University Bloomberg School of Public Health, 615 N Wolfe St, Baltimore, MD 21205, USA; hji@jhu.edu; 8Stanley Manne Children’s Research Institute, Mary Ann & J. Milburn Smith Child Health Research, Outreach and Advocacy Center, Ann & Robert H. Lurie Children’s Hospital of Chicago, 2430 N Halsted St, Chicago, IL 60614, USA; TBartell@luriechildrens.org; 9Integrated Research Center for Fetal Medicine, Department of Gynecology and Obstetrics, Johns Hopkins University School of Medicine, 1800 Orleans St, Baltimore, MD 21287, USA; iburd@jhmi.edu; 10Division of General Pediatrics & Adolescent Medicine, Department of Pediatrics, Johns Hopkins University School of Medicine, 1800 Orleans St, Baltimore, MD 21287, USA

**Keywords:** ADHD, acetaminophen, pregnancy

## Abstract

Previous studies have suggested a positive association between self-reported maternal acetaminophen use during pregnancy and risk of attention deficit hyperactivity disorder (ADHD) in offspring. We sought to examine the prospective association between maternal plasma biomarkers of acetaminophen intake and ADHD diagnosis in the offspring. This report analyzed 1180 children enrolled at birth and followed prospectively as part of the Boston Birth Cohort, including 188 with ADHD diagnosis based on electronic medical record review. Maternal biomarkers of acetaminophen intake were measured in plasma samples obtained within 1–3 days postpartum. Odds ratios for having ADHD diagnosis or other developmental disorders were estimated using multinomial logistic regression models, adjusting for pertinent covariables. Compared to neurotypical children, we observed significant positive dose-responsive associations with ADHD diagnosis for each maternal acetaminophen biomarker. These dose–responsive associations persisted after adjusting for indication of acetaminophen use and other pertinent covariates; and were specific to ADHD, rather than other neurodevelopmental disorders. In the stratified analyses, differential point estimates of the associations were observed across some strata of covariates. However, these differences were not statistically significant. Maternal acetaminophen biomarkers were specifically associated with increased risk of ADHD diagnosis in offspring. Additional clinical and mechanistic investigations are warranted.

## 1. Introduction

Attention deficit hyperactivity disorder (ADHD) is one of the most common lifelong neurodevelopmental disorders in the world. Its prevalence among children ages 4–17 years in the U.S. increased significantly from 7.0% to 10.2% during the past two decades [1,2]. The rapid rise of ADHD cannot be attributed to genetic mutations. Indeed, multiple social and environmental risk factors have been associated with the development of ADHD, including family-related factors [3,4,5,6,7,8,9,10,11,12,13,14,15], maternal obesity [16,17], maternal smoking [8,18,19,20], maternal drinking [8,21], low birthweight and preterm birth [22], exposure to organophosphates [23], polychlorinated biphenyls [24,25], and lead exposure [24,26,27,28,29]. These findings underscore the role of environmental factors in the etiology of ADHD, and the need to explore other important yet unknown risk factors for ADHD [30]. Acetaminophen is widely used and recommended over-the-counter medication for fever and pain relief during pregnancy. The extent of acetaminophen use during pregnancy is over 65% in the U.S. and over 50% in Europe [31,32]. The inhibition of prostaglandin synthesis is part of the therapeutic effect of acetaminophen [33]. Prostaglandins not only act as fever determinants but also play essential roles in brain function, including long-term potentiation [34], learning [35], and cerebellar development [33]. Because of its widespread use and role in brain function, the potential unknown adverse effects of acetaminophen use on developing fetal brain need to be clarified [36].

Since 2013, research studies analyzing five prospective cohorts from Europe and New Zealand have consistently shown a positive association between maternal intake of acetaminophen during pregnancy and increased risk of ADHD and its related symptoms in offspring [37,38,39,40,41]. The Society for Maternal-Fetal Medicine and the Food and Drug Administration expressed concern that the data from these recent studies are still too inconclusive to draw any causal inference between prenatal acetaminophen use and ADHD development in offspring [42,43]. Their primary criticisms included the use of self-reported exposure, lack of dose quantification, and unmeasured confounders [42]. To address the concerns and criticisms related to previous studies and improve our understanding of acetaminophen’s effect during pregnancy and beyond, there is a need for a well-designed prospective birth cohort study with blood samples available to measure maternal acetaminophen levels. Currently, no such study exists.

In this study, using the data from the Boston Birth Cohort, we sought to examine the prospective association between maternal plasma acetaminophen metabolites levels measured within a few days after delivery and ADHD diagnosis in the offspring. Given the timing of our biomarker measurement, we are poised to address a specific question that has not been explored in the previous study: Is the maternal use of acetaminophen during peripartum period (as reflected in plasma biomarker levels) associated with increased risk of ADHD diagnosis in offspring?

## 2. Methods

### 2.1. Sample

Initiated in 1998, mother/infant pairs were recruited at birth from the Boston Medical Center (BMC) for participation in the Boston Birth Cohort (BBC) [44,45]. The BMC serves a predominately low income, urban, minority population and is also the largest safety net hospital in New England. Eligible mothers were those who delivered a singleton live birth at BMC. They were approached for consent and enrollment within 24–72 h after delivery. Infants who continued to receive pediatric primary or specialty care at BMC were invited (beginning at age six months) to participate in the follow-up study in which they are prospectively followed from birth onwards [44,46,47]. After obtaining informed consent, a standardized questionnaire was administered by trained research staff, and a maternal venous blood sample was obtained. Mothers who conceived via in vitro fertilization, multiple-gestation pregnancies, deliveries induced by maternal trauma, and/or newborns with substantial congenital disabilities were not eligible for participation. Both the baseline study and the follow-up study have been approved by the Institutional Review Boards (IRB) of Boston University Medical Center and Johns Hopkins Bloomberg School of Public Health.

As illustrated in the study flowchart, of the 3098 children in the postnatal follow-up study in the BBC, 1412 mothers had sufficient plasma samples for metabolomic assay. Of the mothers with metabolomic data, we further excluded 232 participants who had missing data for key covariates. Our final sample comprised 1180 mother/infant pairs with all pertinent data: exposure, outcome, and covariates (Appendix A). Although this is a subset of the BBC, this sample was similar to the excluded sample in terms of baseline maternal and newborn characteristics (Appendix A), except for having a slightly higher percentage of black children, longer gestation, and higher birthweight.

### 2.2. Definitions for ADHD, ASD, Other DD, and Neurotypical Children

We extracted information regarding each child’s neuro-developmental diagnoses as documented in their electronic medical records (EMRs). Beginning in 2003, BMC implemented EMR as part of routine data collection for both well-child and specialty clinical visits. The primary and secondary diagnoses for each clinical visit were coded in the EMR using the International Classification of Diseases, Ninth Revision (ICD-9) (before 1 October 2015) and ICD-10 (after 1 October 2015). Thus, all children in the BBC postnatal follow-up study with a related ICD-9 (314.0, 314.00, 314.01, 314.1, 314.2, 314.8, or 314.9) or ICD-10 (F90.0, F90.1, F90.2, F90.8, or F90.9) code included in their EMR between 2003 and 2016 were classified as having ADHD. Similarly, children with an ICD-9 (299.0, 299.00, 299.01, 299.8, 299.80, 299.81, 299.9, 299.90, or 299.91) or ICD-10 (F84.0, F84.8, or F84.9) code were classified as having an autism spectrum disorder (ASD). Furthermore, children with any of following developmental disorder diagnoses noted in their EMR were classified as having other developmental disorders (other DD): developmental delays, or intellectual disabilities. Children without any diagnosis of ASD, ADHD, developmental delays, or intellectual disabilities were classified as neurotypical (NT). Appendix A lists the ICD-9 and ICD-10 codes for each developmental disorder diagnosis.

### 2.3. Maternal Biomarkers of Acetaminophen Use

Maternal plasma biomarkers of acetaminophen use were measured using nonfasting blood samples obtained within 1–3 days postpartum. As illustrated in Figure 1, the main metabolites (and proportion) of acetaminophen include unchanged acetaminophen (~5%), acetaminophen glucuronide (52–57%), acetaminophen sulfate (30–44%), and hepatotoxic N-acetyl-p-benzoquinone imine (NAPQI) (5–10%). NAPQI can be further detoxified as 3-(*N*-Acetyl-l-cystein-*S*-yl) acetaminophen [48]. The peak intensity of unchanged acetaminophen, acetaminophen glucuronide, and 3-(*N*-Acetyl-l-cystein-*S*-yl) acetaminophen in maternal blood was measured using liquid chromatography-tandem mass spectrometry (LC-MS) techniques at the MIT Broad Institute Metabolite Profiling Laboratory. All the intensity levels were inverse normal transformed for the subsequent statistical analyses.

### 2.4. Covariates

Based on previous literature [29,37,38,39,40,41], the following covariates were included as potential confounders: maternal age at delivery, maternal race/ethnicity, maternal education, smoking from six months before pregnancy to birth (never smoked, quit during this period, continued to smoke during this period), ever drank alcohol from six months before pregnancy to birth, maternal pre-pregnancy BMI, parity, maternal fever during pregnancy, intrauterine infection/inflammation, baby’s sex, delivery type, gestational age, birthweight, breastfeeding, early childhood lead levels, and maternal high-density lipoprotein (HDL) levels. Maternal demographic covariates were collected using a standard questionnaire interview. Maternal and child clinically-related covariates were abstracted from their medical records, respectively. The lead levels of the children were collected as part of the pediatric routine lead screening and extracted from their EMRs. The first lead levels measured were chosen for the analysis. Maternal plasma HDL levels were measured using nonfasting blood samples obtained within 1–3 days postpartum.

### 2.5. Statistical Analyses

The characteristics of the study sample for the ADHD, ASD (excluding participants with ADHD diagnosis), other DD, and NT groups were compared using one-way ANOVA for continuous variables and χ^2^ tests for categorical variables. The main exposures analyzed in this study were maternal acetaminophen metabolite levels, which were inverse normal transformed to approximate the normal distribution. The inverse normal transformed unchanged acetaminophen levels were also categorized into tertiles. Due to the high rate of non-detection, the inverse normal transformed acetaminophen glucuronide and 3-(*N*-Acetyl-l-cystein-*S*-yl) acetaminophen levels were categorized into three groups: no detection, below median, above median of detected values. Based on previous findings regarding the proportions of acetaminophen metabolites typically found in blood samples [48], we further calculated a variable to reflect overall “acetaminophen burden” by combining all of the acetaminophen metabolites levels with a weighting of their proportions in the acetaminophen metabolic pathway (acetaminophen burden = (unchanged acetaminophen/5%+ acetaminophen glucuronide/50%+ 3-(*N*-Acetyl-l-cystein-*S*-yl) acetaminophen/5%)/60%) [48]. The acetaminophen burden levels were then also categorized into three groups: no detection, below median, and above median. Each child’s early life lead level was converted into a binary variable (5 µg/dL as the cutoff) for analysis based on CDC guidelines [49]. Maternal HDL level was cut at 60 mg/dL for analyses based on previous finding [50].

We conducted sequential multinomial logistic regression models to examine the association between maternal acetaminophen metabolite levels and the risk of having ADHD diagnosis, ASD diagnosis (excluding ADHD diagnosis), or other DD diagnosis in offspring. Multinomial logistic regression is a method that generalizes logistic regression to the tests with more than two possible discrete outcomes [51]. Except for the fact that dependent variable is categorical rather than binary, the basic setup is similar to logistic regression [51]. Models included a crude (unadjusted) model (Model 1); a model adjusted for maternal age at delivery, maternal race/ethnicity, maternal education, smoking during pregnancy, drinking during pregnancy, parity, maternal pre-pregnancy BMI, baby’s sex, delivery type, gestational age, and birthweight (Model 2); and models further adjusted for maternal fever during pregnancy (Model 3), intrauterine infection/inflammation (Model 4), and breastfeeding (Model 5), separately, and combined (Model 6). We also performed stratified analyses by each stratum of covariates (including child’s early life lead levels [52] and maternal HDL levels [50]) for binary acetaminophen burden (detected vs. no detection) using univariate logistic regression comparing those with an ADHD diagnosis to the NT group. For the sensitivity analyses, we repeated the sequential models using propensity score weighted multinomial logistic regression. The propensity score was calculated based on all the covariates in Model 6 using *psmatch2* package. We further repeated the sequential models for each of the following outcomes: “ADHD only” (excluding ASD diagnosis), “ASD only” (excluding ADHD diagnosis), and “ADHD and ASD” (having both diagnoses), all compared to the NT group. STATA^®^ version 14.0 software was used to perform all analyses (Stata Corporation, College Station, TX, USA).

## 3. Results

In the final sample, there were 188 children with a diagnosis of ADHD, 44 children with a diagnosis of ASD (without ADHD diagnosis), 344 children with a diagnosis of other DD, and 604 NT children. The median age at first ADHD diagnosis was 7 years. Figure 2 shows the distribution of each acetaminophen metabolite and acetaminophen burden across diagnosis groups. Both the ADHD diagnosis and ASD diagnosis (without ADHD diagnosis) groups had more mothers with higher levels of acetaminophen metabolites compared to the NT and other DD diagnosis groups. Table 1 presents the crude comparisons of maternal and child characteristics among the ADHD diagnosis, ASD diagnosis (without ADHD diagnosis), other DD diagnosis, and NT groups. The ADHD and ASD groups had the highest percentage of detectable unchanged acetaminophen and its metabolites. Mothers of children with any ADHD diagnosis were also more likely to have below college degree education, ever smoked before or during pregnancy, and C-section delivery, compared with the NT group. Children with any ADHD, ASD, or any other DD diagnosis were more likely to be male, born prematurely and have had low birthweight, compared with the NT group. 

Table 2 shows the sequential multinomial logistic regression model results for the relationship between acetaminophen metabolites and the risk of ADHD diagnosis, ASD diagnosis (excluding ADHD), or other DD diagnosis, and before and after adjusting for pertinent covariates. The group with the highest plasma level of each acetaminophen metabolite was significantly associated with the risk of ADHD diagnosis, and the effect size was similar across all models. Moreover, we identified dose-responsive patterns across all acetaminophen metabolites and burden. Compared to levels in the non-detection category, below median and above median levels of maternal acetaminophen burden were associated with a 58% and 88% increase in the odds of ADHD diagnosis, respectively (Model 6: OR for below median = 1.58, 95% CI (1.02, 2.46); OR for above median = 1.88, 95% CI (1.18, 3.00)). In contrast, the risks of ASD diagnosis and other DD diagnoses were not significantly associated with maternal plasma levels of acetaminophen metabolites across all models. Appendix A further confirms that in our sensitivity analyses the acetaminophen metabolite levels were specifically associated with the risk of having an ADHD diagnosis (without ASD diagnosis). Appendix A presents the results of propensity score weighted multinomial logistic regression models, which is an additional method of controlling for unmeasured confounders. It also shows similar findings to Table 2.

We also explored if the associations between acetaminophen metabolites and ADHD varied by strata of covariables. Figure 3 presents the forest plot of the stratified analyses for binary acetaminophen burden (detected vs. non-detection) by each stratum of covariates using simple logistic regression comparing ADHD diagnosis to NT. The point estimates of the acetaminophen burden-ADHD associations were similar among strata of maternal age, smoking before or during pregnancy, maternal obesity, and early life lead exposure. On the other hand, a larger difference in the point estimate of the odds ratios was observed across strata of child’s sex, parity, intrauterine infection/inflammation, alcohol drinking before or during pregnancy, delivery type, birthweight, gestational age, breastfeeding, and maternal HDL. However, tests of interaction between each covariate and binary acetaminophen burden (detected vs. non-detection) were not significant.

## 4. Discussion

In this prospective birth cohort study, we found a significant positive association between maternal blood acetaminophen metabolite levels measured within 1–3 days postpartum and ADHD diagnosis in offspring; such an association was not observed for other developmental disorders. This association remained even after adjusting for indication factors of acetaminophen use (maternal fever and maternal intrauterine infection/inflammation during pregnancy) and other pertinent covariates. This study has contributed the following new information to the field.

Even though positive associations between maternal reported intake of acetaminophen during pregnancy and risk of ADHD diagnosis in their offspring have been reported by multiple independent large cohort studies [37,38,39,40,41], there has been a dearth of prospective birth cohort studies to examine objective biomarkers of acetaminophen use to address specific concerns about self-reported exposure and lack of dose quantification in those studies.

To our knowledge, this is the first prospective birth cohort study to examine the association between objective maternal plasma biomarkers of acetaminophen and offspring ADHD diagnosis, and to take into account a large number of potential covariables. Our study was further strengthened by the diagnosis of ADHD by both general pediatricians and developmental specialists. By demonstrating a prospective and dose–response relationship using biomarkers specific to acetaminophen intake during early postpartum, our study findings further extend the previously identified positive association between acetaminophen and ADHD from pregnancy into postpartum.

Although the causality and biological mechanisms underlying the maternal acetaminophen and child ADHD association remain to be determined, the potential for neurotoxicity is plausible according to previous findings. First, the therapeutic effect of acetaminophen inhibits prostaglandin production [33]. Prostaglandin synthesis involves multiple essential biological processes underlying the function and development of the brain, such as long-term potentiation [34], learning [35], and cerebellar development [33]. Second, accumulating studies have shown that acetaminophen not only rapidly enters the cerebrospinal fluid but also shows a profound influence on adult brain function [53,54,55,56]. Third, maternal acetaminophen can be readily transferred to the fetus through placenta [57] and to the infant via breastfeeding [58,59,60]. Acetaminophen remains in fetal/infant circulation much longer than it does in adults [57]. The maternal usage of phenacetin (could be converted into acetaminophen rapidly in adults) 5.5 h before delivery could lead to detectable acetaminophen metabolites in infant’s urine until the first 26 h of life [57]. The prolonged detection of acetaminophen among children is due to their undeveloped liver, which slowly metabolizes the acetaminophen [61]. On the one hand, the low metabolic capacity in early life makes it safer for children to use acetaminophen because of slower production of toxic NAPQI, but, on the other hand, it makes the fetus/infant more vulnerable to maternal metabolized toxic NAPQI during pregnancy and breastfeeding. Our analysis stratified by breastfeeding history shows stronger associations among women who ever breastfeed their babies, which provides indirect support for the impact of acetaminophen through lactation. It is possible that fetal/infant exposure to maternal acetaminophen metabolites via placenta or breastfeeding, coupled with limited metabolic capacity, might lead to both direct toxic damage from maternal NAPQI and potential disruption in neurodevelopment function due to prostaglandin inhibition.

While tests of interaction were not significant (likely due to lack of power), our stratified analyses identified multiple maternal and fetal factors that may enhance the association between maternal acetaminophen metabolites and ADHD diagnosis in offspring. These potential effect modifiers are biologically plausible and warrant further investigations. For instance, we found that the effect size of acetaminophen use on the risk of ADHD diagnosis is more pronounced among women who drank alcohol six months before or during pregnancy. Effect modification by alcohol is supported by biological studies [62,63]. A mechanistic study showed that ethanol could cause induction of cytochrome P450 2E1 and selective depletion of mitochondrial glutathione, which could lead to limited clearance capacity of the toxic NAPQI [63]. Additionally, the stronger and more significant acetaminophen–ADHD association among female children indicates the need to further investigate the potential sex-specific biological mechanism underlying the acetaminophen exposure.

Our study also had some limitations. First, this study only included a one-time measurement of maternal acetaminophen metabolite levels within 1–3 days postpartum. Acetaminophen has a 2–3 h half-life in adults [64], thus the detectable levels of acetaminophen biomarkers can only reflect women’s recent use, which could be influenced by both dosage and metabolism capacity. As a result, we cannot pinpoint the exact dosage and usage pattern at specific time periods (pre-conception, specific trimester during pregnancy, perinatal and postnatal period) based on this one-time measurement. Nonetheless, women with detectable levels of acetaminophen biomarkers are likely to be more regular users. The findings would be strengthened if we could have included maternal acetaminophen metabolite measures taken at least once for each trimester as well as cord blood biomarkers. Given the fact that the rate of prenatal acetaminophen use during pregnancy is over 65% in the U.S. [32], the one-time measurement in our study likely reflects maternal acetaminophen use during the peripartum period. Our future study with cord blood metabolite measurements will provide further confirmation. Second, our metabolite measurement method did not include acetaminophen sulfate, which accounts for 30–44% of the total metabolites of acetaminophen under the normal dosage [48]. Fortunately, the total acetaminophen burden can be estimated based on the relatively stable proportions of other measured acetaminophen metabolites [48]. Third, although we adjusted for major known risk factors of ADHD and indication of acetaminophen use, we could not adjust for several familial factors identified in previous studies [3,4,5,13,14,15,65]. We also cannot rule out the possibility of unmeasured or unknown residual confounding, although our propensity score analyses seek to adjust for such confounding and provide additional credibility for our findings. Lastly, our study sample consists of a predominantly urban low-income minority population. This characteristic may limit the generalization of our results to all pregnant women living in the U.S.

## 5. Conclusions

Maternal plasma biomarkers of acetaminophen use measured within a few days after delivery were specifically associated with increased risk of ADHD diagnosis in offspring, not with other developmental disorders. This association remained after adjusting for multiple previously identified potential confounders and potential indications for acetaminophen use. While our study provides the first objective biomarker evidence of the relationship between maternal acetaminophen use and ADHD diagnosis in offspring, we could not provide definitive support for a causal inference of this relationship, given the observational nature of this study and the limitations outlined above. However, our evidence drawn from the perinatal period lends further support for the association between maternal acetaminophen use and ADHD in offspring, which has been illustrated by multiple large studies of acetaminophen use during pregnancy via self-reported information. If further confirmed, these results may help address concerns raised by the Society for Maternal-Fetal Medicine and the U.S. Food and Drug Administration [42,43]. Taking past findings together with the findings from this study, the potential adverse effect of maternal acetaminophen use on ADHD risk in offspring warrants additional investigations.

## Figures and Tables

**Figure 1 brainsci-08-00127-f001:**
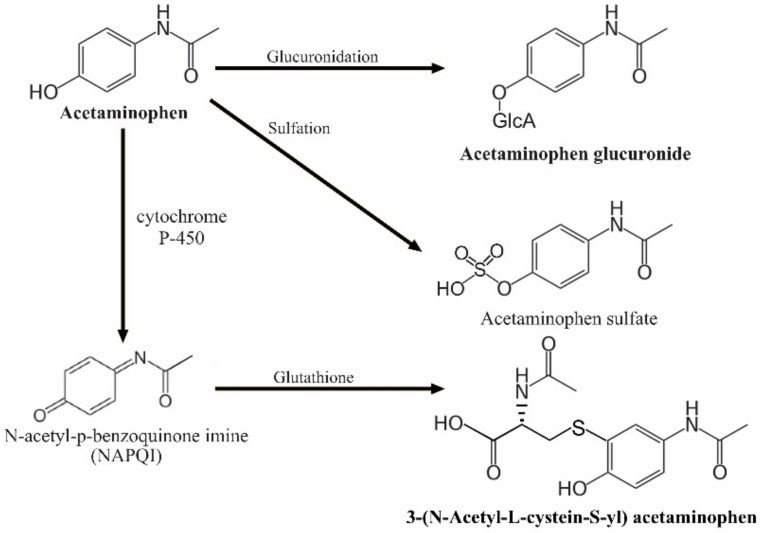
Pathways of acetaminophen metabolism (Bolded metabolites were measured in this study).

**Figure 2 brainsci-08-00127-f002:**
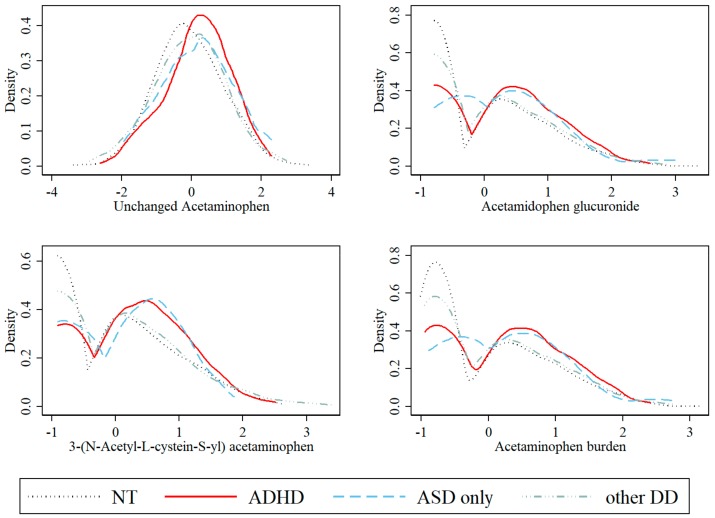
Comparison of the distributions of acetaminophen metabolites and acetaminophen burden by specific diagnosis groups. NT was defined as free of any developmental disorder diagnosis; ADHD was defined as any ADHD diagnosis; ASD only was defined as any ASD diagnosis without having an ADHD diagnosis; other DD was defined as any developmental disorder diagnosis other than ASD and ADHD.

**Figure 3 brainsci-08-00127-f003:**
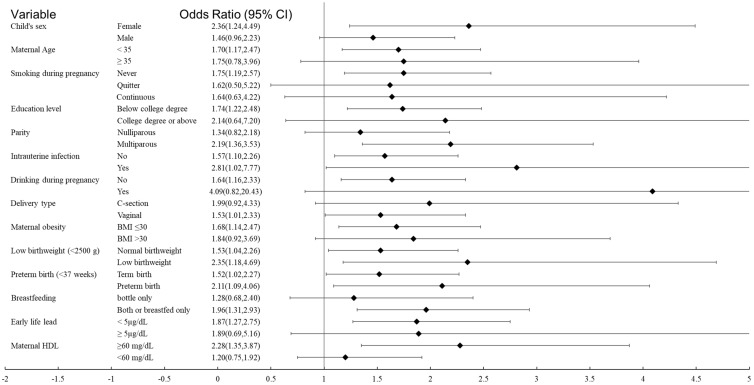
The forest plot for the crude association between maternal binary acetaminophen burden and the risk of ADHD diagnosis in offspring across each stratum of pertinent covariables. Acetaminophen burden is the sum of all the acetaminophen metabolites. No detection group is the reference group.

**Table 1 brainsci-08-00127-t001:** Maternal and child characteristics for children with ADHD diagnosis, ASD diagnosis (excluding ADHD), other developmental disorder diagnosis (other DD), and neurotypical children (NT).

Variable	Total, *N* (%)	NT, *N* (%)	ADHD, *N* (%)	ASD, *N* (%)	Other DD, *N* (%)	*p*-Value ^ǂ^
Total	1180 (100)	604 (51.19)	188 (15.93)	44 (3.73)	344 (29.15)	
Maternal Age						0.101
<35	965 (81.78)	510 (84.44)	151 (80.32)	34 (77.27)	270 (78.49)	
≥35	215 (18.22)	94 (15.56)	37 (19.68)	10 (22.73)	74 (21.51)	
Maternal race/ethnicity						0.073
Black	809 (68.56)	425 (70.36)	126 (67.02)	26 (59.09)	232 (67.44)	
White	48 (4.07)	24 (3.97)	11 (5.85)	2 (4.55)	11 (3.20)	
Hispanic	256 (21.69)	112 (18.54)	44 (23.40)	15 (34.09)	85 (24.71)	
Others	67 (5.68)	43 (7.12)	7 (3.72)	1 (2.27)	16 (4.65)	
Education level						0.208
Below college degree	1033 (87.54)	520 (86.09)	172 (91.49)	37 (84.09)	304 (88.37)	
College degree or above	147 (12.46)	84 (13.91)	16 (8.51)	7 (15.91)	40 (11.63)	
Smoking before or during pregnancy						0.018
Never	977 (82.80)	520 (86.09)	141 (75.00)	38 (86.36)	278 (80.81)	
Quitter	90 (7.63)	38 (6.29)	18 (9.57)	3 (6.82)	31 (9.01)	
Continuous	113 (9.58)	46 (7.62)	29 (15.43)	3 (6.82)	35 (10.17)	
Drinking before or during pregnancy						0.491
No	1086 (92.03)	560 (92.72)	173 (92.02)	38 (86.36)	315 (91.57)	
Yes	94 (7.97)	44 (7.28)	15 (7.98)	6 (13.64)	29 (8.43)	
Parity						0.484
Nulliparous	527 (44.66)	281 (46.52)	85 (45.21)	18 (40.91)	143 (41.57)	
Multiparous	653 (55.34)	323 (53.48)	103 (54.79)	26 (59.09)	201 (58.43)	
Child’s sex						<0.001
Female	576 (48.81)	351 (58.11)	49 (26.06)	14 (31.82)	162 (47.09)	
Male	604 (51.19)	253 (41.89)	139 (73.94)	30 (68.18)	182 (52.91)	
Delivery type						0.008
C-section	426 (36.10)	192 (31.79)	75 (39.89)	22 (50.00)	137 (39.83)	
Vaginal	754 (63.90)	412 (68.21)	113 (60.11)	22 (50.00)	207 (60.17)	
Maternal fever						0.594
No	1108 (93.90)	570 (94.37)	178 (94.68)	42 (95.45)	318 (92.44)	
Yes	72 (6.10)	34 (5.63)	10 (5.32)	2 (4.55)	26 (7.56)	
Intrauterine infection/inflammation						0.136
No	1023 (86.69)	537 (88.91)	157 (83.51)	38 (86.36)	291 (84.59)	
Yes	157 (13.31)	67 (11.09)	31 (16.49)	6 (13.64)	53 (15.41)	
Maternal BMI						0.304
<18.50	41 (3.47)	20 (3.31)	9 (4.79)	2 (4.55)	10 (2.91)	
18.50–24.99	514 (43.56)	284 (47.02)	72 (38.30)	15 (34.09)	143 (41.57)	
25–29.99	337 (28.56)	164 (27.15)	58 (30.85)	11 (25.00)	104 (30.23)	
>30	288 (24.41)	136 (22.52)	49 (26.06)	16 (36.36)	87 (25.29)	
Breastfeeding						0.351
Bottle only	286 (24.24)	142 (23.51)	55 (29.26)	9 (20.45)	80 (23.26)	
Both or breastfed only	894 (75.76)	462 (76.49)	133 (70.74)	35 (79.55)	264 (76.74)	
Unchanged acetaminophen *						0.027
First tertile	411 (34.83)	227 (37.58)	46 (24.47)	13 (29.55)	125 (36.34)	
Second tertile	375 (31.78)	192 (31.79)	66 (35.11)	12 (27.27)	105 (30.52)	
Third tertile	394 (33.39)	185 (30.63)	76 (40.43)	19 (43.18)	114 (33.14)	
3-(*N*-Acetyl-l-cystein-*S*-yl) acetaminophen *						0.013
No detection	441 (37.37)	248 (41.06)	51 (27.13)	15 (34.09)	127 (36.92)	
Below median	361 (30.59)	182 (30.13)	62 (32.98)	10 (22.73)	107 (31.10)	
Above median	378 (32.03)	174 (28.81)	75 (39.89)	19 (43.18)	110 (31.98)	
Acetaminophen glucuronide *						0.018
No detection	531 (45.00)	299 (49.50)	68 (36.17)	15 (34.09)	149 (43.31)	
Below median	315 (26.69)	152 (25.17)	52 (27.66)	15 (34.09)	96 (27.91)	
Above median	334 (28.31)	153 (25.33)	68 (36.17)	14 (31.82)	99 (28.78)	
Acetaminophen burden **						0.027
No detection	531 (45.00)	299 (49.50)	68 (36.17)	15 (34.09)	149 (43.31)	
Below median	315 (26.69)	151 (25.00)	54 (28.72)	14 (31.82)	96 (27.91)	
Above median	334 (28.31)	154 (25.50)	66 (35.11)	15 (34.09)	99 (28.78)	
Gestational age, week						<0.001
Mean (SD)	37.9 (3.3)	38.5 (2.5)	37.3 (3.6)	37.0 (4.6)	37.2 (4.0)	
Birthweight, g						<0.001
Mean (SD)	2966.2 (789.9)	3085.5 (669.7)	2865.0 (819.4)	2860.9 (1026.2)	2825.5 (898.0)	

Note: NT was defined as free of any developmental disorder diagnosis; ADHD was defined as any ADHD diagnosis; ASD was defined as any ASD diagnosis without having an ADHD diagnosis; other DD was defined as any developmental disorder diagnosis other than ASD and ADHD. ^ǂ^ The *p*-values were obtained from χ^2^ tests or one-way ANOVA among the four diagnosis groups; * Inverse normal transformed intensity; ** Sum of all the acetaminophen metabolites.

**Table 2 brainsci-08-00127-t002:** The association between maternal acetaminophen metabolites and the risk of ADHD diagnosis, ASD diagnosis (excluding ADHD), and other DD diagnosis in offspring.

Model		ADHD, 188 (15.9%)			ASD, 44 (3.7%)			Other DD, 344 (29.2%)		
		Odds Ratio	95% CI	*p*-Value	Odds Ratio	95% CI	*p*-Value	Odds Ratio	95% CI	*p*-Value
Unchanged acetaminophen *										
Model 1	Second tertile	1.70	(1.11,2.59)	0.014	1.09	(0.49,2.45)	0.832	0.99	(0.72,1.37)	0.967
	Third tertile	2.03	(1.34,3.07)	0.001	1.79	(0.86,3.73)	0.118	1.12	(0.81,1.54)	0.490
Model 2	Second tertile	1.72	(1.10,2.70)	0.018	0.98	(0.43,2.27)	0.970	0.99	(0.71,1.40)	0.977
	Third tertile	2.08	(1.29,3.35)	0.003	1.38	(0.60,3.18)	0.451	0.94	(0.65,1.35)	0.732
Model 3	Second tertile	1.73	(1.10,2.72)	0.017	0.99	(0.43,2.30)	0.989	0.97	(0.69,1.37)	0.883
	Third tertile	2.08	(1.29,3.35)	0.003	1.39	(0.60,3.20)	0.443	0.93	(0.65,1.35)	0.706
Model 4	Second tertile	1.71	(1.09,2.68)	0.020	0.98	(0.42,2.27)	0.968	0.98	(0.70,1.39)	0.931
	Third tertile	2.06	(1.28,3.33)	0.003	1.38	(0.60,3.18)	0.453	0.93	(0.65,1.35)	0.705
Model 5	Second tertile	1.72	(1.10,2.70)	0.018	0.98	(0.42,2.26)	0.961	0.99	(0.70,1.39)	0.958
	Third tertile	2.06	(1.28,3.32)	0.003	1.40	(0.61,3.23)	0.432	0.94	(0.65,1.36)	0.749
Model 6	Second tertile	1.74	(1.10,2.73)	0.017	0.99	(0.43,2.29)	0.979	0.97	(0.69,1.37)	0.869
	Third tertile	2.05	(1.27,3.32)	0.003	1.40	(0.60,3.24)	0.433	0.93	(0.65,1.35)	0.718
3-(*N*-Acetyl-l-cystein-*S*-yl) acetaminophen *										
Model 1	Below median	1.66	(1.09,2.51)	0.018	0.91	(0.40,2.07)	0.819	1.15	(0.83,1.58)	0.399
	Above median	2.10	(1.40,3.14)	<0.001	1.81	(0.89,3.65)	0.100	1.23	(0.90,1.70)	0.198
Model 2	Below median	1.68	(1.08,2.61)	0.021	0.73	(0.31,1.72)	0.474	1.08	(0.77,1.52)	0.644
	Above median	2.06	(1.28,3.31)	0.003	1.21	(0.53,2.75)	0.653	0.96	(0.66,1.40)	0.835
Model 3	Below median	1.70	(1.09,2.65)	0.020	0.74	(0.31,1.75)	0.494	1.05	(0.75,1.48)	0.763
	Above median	2.06	(1.28,3.31)	0.003	1.22	(0.53,2.78)	0.640	0.95	(0.65,1.38)	0.789
Model 4	Below median	1.66	(1.06,2.58)	0.025	0.73	(0.31,1.72)	0.468	1.06	(0.76,1.49)	0.716
	Above median	2.04	(1.27,3.28)	0.003	1.20	(0.53,2.75)	0.661	0.95	(0.65,1.38)	0.785
Model 5	Below median	1.67	(1.07,2.60)	0.024	0.74	(0.31,1.75)	0.497	1.09	(0.78,1.53)	0.619
	Above median	2.04	(1.27,3.28)	0.003	1.23	(0.54,2.82)	0.621	0.97	(0.67,1.41)	0.868
Model 6	Below median	1.68	(1.08,2.63)	0.022	0.75	(0.32,1.78)	0.513	1.06	(0.75,1.49)	0.734
	Above median	2.03	(1.26,3.27)	0.004	1.23	(0.54,2.82)	0.626	0.96	(0.66,1.39)	0.811
Acetaminophen glucuronide *										
Model 1	Below median	1.50	(1.00,2.27)	0.051	1.97	(0.94,4.13)	0.074	1.27	(0.92,1.75)	0.150
	Above median	1.95	(1.33,2.88)	0.001	1.82	(0.86,3.88)	0.118	1.30	(0.94,1.79)	0.110
Model 2	Below median	1.49	(0.96,2.31)	0.074	1.47	(0.68,3.19)	0.332	1.14	(0.81,1.60)	0.465
	Above median	2.03	(1.28,3.22)	0.003	1.26	(0.53,2.99)	0.602	1.07	(0.73,1.55)	0.738
Model 3	Below median	1.51	(0.97,2.34)	0.068	1.50	(0.69,3.29)	0.306	1.11	(0.78,1.56)	0.569
	Above median	2.03	(1.28,3.23)	0.003	1.28	(0.54,3.04)	0.579	1.05	(0.72,1.53)	0.787
Model 4	Below median	1.47	(0.95,2.29)	0.085	1.47	(0.67,3.21)	0.333	1.12	(0.80,1.58)	0.516
	Above median	2.01	(1.26,3.18)	0.003	1.26	(0.53,3.00)	0.599	1.05	(0.73,1.53)	0.784
Model 5	Below median	1.49	(0.96,2.31)	0.075	1.47	(0.68,3.20)	0.330	1.13	(0.81,1.59)	0.466
	Above median	2.01	(1.27,3.19)	0.003	1.27	(0.54,3.02)	0.584	1.07	(0.74,1.55)	0.726
Model 6	Below median	1.51	(0.97,2.35)	0.068	1.50	(0.69,3.28)	0.308	1.11	(0.78,1.56)	0.564
	Above median	2.00	(1.26,3.18)	0.003	1.28	(0.54,3.05)	0.578	1.05	(0.73,1.53)	0.780
Acetaminophen burden **										
Model 1	Below median	1.57	(1.05,2.36)	0.029	1.85	(0.87,3.93)	0.110	1.28	(0.92,1.76)	0.139
	Above median	1.88	(1.28,2.78)	0.001	1.94	(0.92,4.08)	0.080	1.29	(0.94,1.78)	0.119
Model 2	Below median	1.56	(1.01,2.42)	0.045	1.39	(0.63,3.06)	0.410	1.14	(0.81,1.61)	0.439
	Above median	1.91	(1.21,3.04)	0.006	1.36	(0.58,3.20)	0.477	1.05	(0.73,1.53)	0.779
Model 3	Below median	1.58	(1.02,2.45)	0.041	1.43	(0.64,3.15)	0.381	1.11	(0.79,1.57)	0.543
	Above median	1.92	(1.21,3.05)	0.006	1.38	(0.59,3.25)	0.457	1.04	(0.72,1.52)	0.824
Model 4	Below median	1.54	(1.00,2.39)	0.052	1.39	(0.63,3.08)	0.411	1.13	(0.80,1.59)	0.488
	Above median	1.89	(1.19,3.01)	0.007	1.37	(0.58,3.22)	0.475	1.04	(0.72,1.51)	0.826
Model 5	Below median	1.56	(1.01,2.41)	0.045	1.39	(0.63,3.07)	0.409	1.14	(0.81,1.61)	0.442
	Above median	1.90	(1.20,3.02)	0.007	1.38	(0.59,3.25)	0.459	1.06	(0.73,1.54)	0.763
Model 6	Below median	1.58	(1.02,2.46)	0.040	1.43	(0.64,3.15)	0.382	1.11	(0.79,1.57)	0.539
	Above median	1.88	(1.18,3.00)	0.008	1.39	(0.59,3.27)	0.456	1.05	(0.72,1.52)	0.815

Note: NT was defined as free of any developmental disorder diagnosis; ADHD was defined as any ADHD diagnosis; ASD was defined as any ASD diagnosis without having an ADHD diagnosis; other DD was defined as any developmental disorder diagnosis other than ASD and ADHD; Model 1: Multinomial logistic regression without adjustment; Model 2: Model 1 further adjusted for maternal age at delivery, maternal race/ethnicity, maternal education, smoking before or during pregnancy, drinking before or during pregnancy, maternal BMI, parity, child’s sex, delivery type, preterm birth, and birthweight; Model 3: Model 2 further adjusted for maternal fever during pregnancy; Model 4: Model 2 further adjusted for maternal intrauterine infection/inflammation during pregnancy; Model 5: Model 2 further adjusted for breastfeeding; Model 6: Model 2 further adjusted for maternal fever, maternal intrauterine infection/inflammation during pregnancy, and breastfeeding. * Inverse normal transformed intensity ** Sum of all the acetaminophen metabolites. Unchanged acetaminophen: first tertile as reference; For other exposures: no detection as reference.

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
