# Peer review of "Maternal Biomarkers of Acetaminophen Use and Offspring Attention Deficit Hyperactivity Disorder"

_brainsci, 2018, doi:10.3390/brainsci8070127_

Round 1

Reviewer 1 Report

This is an important study. The findings will be of considerable interest.

My major concern is that there is no information on acetaminophen intake between delivery and the blood sample. The authors should be clear on to what extent this was measured, and list it as a limitation. 

The authors sum up a range of social and environmental risk factors associated with ADHD. When possible, they should give particular attention to studies including a contrafactual designs for exposures e.g. (1-3).

In the method section, please provide a description what the different intensity levels could mean in terms of actual use (e.g. How long since last use, dosage).

Only antecedents of the exposure should be included as covariates. Hence, only include breastfeeding before the blood sample was taken and exclude early childhood lead levels (i.e. breastfeeding or lead exposure after the blood sample cannot confound the association).

1.         Chen Q, Sjolander A, Langstrom N, Rodriguez A, Serlachius E, D'Onofrio BM, et al. Maternal pre-pregnancy body mass index and offspring attention deficit hyperactivity disorder: a population-based cohort study using a sibling-comparison design. Int J Epidemiol. 2014;43(1):83-90.

2.         Gustavson K, Ystrom E, Stoltenberg C, Susser E, Suren P, Magnus P, et al. Smoking in Pregnancy and Child ADHD. Pediatrics. 2017;139(2).

3.         Eilertsen EM, Gjerde LC, Reichborn-Kjennerud T, Orstavik RE, Knudsen GP, Stoltenberg C, et al. Maternal alcohol use during pregnancy and offspring attention-deficit hyperactivity disorder (ADHD): a prospective sibling control study. Int J Epidemiol. 2017.

Author Response

Reviewer 1:

This is an important study. The findings will be of considerable interest.

1. My major concern is that there is no information on acetaminophen intake between delivery and the blood sample. The authors should be clear on to what extent this was measured, and list it as a limitation.

Response:

The point is well taken. We don’t have detailed information about acetaminophen intake between delivery and the blood sample. Due to acetaminophen’s 2-3 hours half-life in adults, our one-time measurement can only reflect recent use. This has been more specifically noted in the Limitations. On the other hand, the strong and robust associations with the risk of ADHD diagnosis we observed in this study are worthy of further investigation. It is possible that these women with detectable levels of acetaminophen biomarkers were more likely to be regular users given its short half-life.  We have

2. The authors sum up a range of social and environmental risk factors associated with ADHD. When possible, they should give particular attention to studies including a contrafactual designs for exposures e.g. (1-3).

1.         Chen Q, Sjolander A, Langstrom N, Rodriguez A, Serlachius E, D'Onofrio BM, et al. Maternal pre-pregnancy body mass index and offspring attention deficit hyperactivity disorder: a population-based cohort study using a sibling-comparison design. Int J Epidemiol. 2014;43(1):83-90.

2.         Gustavson K, Ystrom E, Stoltenberg C, Susser E, Suren P, Magnus P, et al. Smoking in Pregnancy and Child ADHD. Pediatrics. 2017;139(2).

3.         Eilertsen EM, Gjerde LC, Reichborn-Kjennerud T, Orstavik RE, Knudsen GP, Stoltenberg C, et al. Maternal alcohol use during pregnancy and offspring attention-deficit hyperactivity disorder (ADHD): a prospective sibling control study. Int J Epidemiol. 2017.

Response:

We would like to clarify that our study is a population-based prospective birth cohort design. The three studies (1-3) reviewer mentioned above applied a sibling-comparison design to adjust for potential unmeasured familial confounding, which is not feasible for our study. Those studies indicated that unmeasured familial confounding could explain those associations. In the revision, we included those studies in the references. In addition, to address the concern regarding unmeasured confounding, we repeated the sequential analysis using propensity score weighted multinomial logistic regression. This method tries to mimic the qualities of a randomized trial in an observational study. The results are consistent with our initial findings (Supplemental Table S4). The associations between acetaminophen biomarkers and risk of ADHD diagnosis are strong and robust for both unadjusted (Model 1) and adjusted models (Model 2-6).

3. In the method section, please provide a description what the different intensity levels could mean in terms of actual use (e.g. How long since last use, dosage).

Response:

Due to acetaminophen’s 2-3 hours half-life in adults, the detectable levels of acetaminophen biomarkers can only reflect the recent use, which could be influenced by dosage but also by each woman’s metabolism.  Consequently, we cannot pinpoint the exact dosage and usage pattern based on this one-time measurement. We acknowledged this limitation in the discussion section.

4. Only antecedents of the exposure should be included as covariates. Hence, only include breastfeeding before the blood sample was taken and exclude early childhood lead levels (i.e. breastfeeding or lead exposure after the blood sample cannot confound the association).

Response:

Our Table 2 listed six sequential models which also included models that did not adjust for breastfeeding and lead level. Even though breastfeeding and lead exposure occur after the blood sample, they are associated with the development of ADHD and not likely in the pathway from acetaminophen use and ADHD. Thus, they may serve as precision variables rather than adjustment for confounding. Our presentation with and without these factors allows readers to examine the effects of inclusion versus exclusion on risk estimates. Presenting multivariable models with and without these covariates allows the readers to decide which multivariable model estimate they prefer.

Reviewer 2 Report

This manuscript reports research on Maternal biomarkers of acetaminophen use and offspring attention deficit hyperactivity disorder. The manuscript is well written and interesting to the readers. However, authors need to improve the manuscript before it is considered for publication.

Some of the comments are given below:

1) Introduction needs to be further improved with current literature

2) More statistical analysis and results need to added

3) Multinomial logistic regression model need to be explained in detail

4) Authors need to compare the results with the other similar methods

4) Conclusion section needs to be further improved.

Author Response

Reviewer 2:

This manuscript reports research on Maternal biomarkers of acetaminophen use and offspring attention deficit hyperactivity disorder. The manuscript is well written and interesting to the readers. However, authors need to improve the manuscript before it is considered for publication.

Some of the comments are given below:

Introduction needs to be further improved with current literature

Response:

We have searched again and added latest relevant references into the introduction.

More statistical analysis and results need to added

Response:

To answer the unmeasured confounding concern, we repeated the sequential analysis using propensity score weighted multinomial logistic regression. The propensity score was calculated based on all the covariates in Model 6 using psmatch2 package. The results showed similar findings to our Table 2. The associations between acetaminophen biomarkers and risk of ADHD diagnosis are strong and robust for both unadjusted (Model 1) and adjusted models (Model 2-6).

Multinomial logistic regression model need to be explained in detail

Response:

We have added more explanation for multinomial logistic regression, along with reference.

Authors need to compare the results with the other similar methods

Response:

To our knowledge, this is the first study to investigate the association between maternal blood acetaminophen metabolite levels measured within 1-3 days postpartum and ADHD diagnosis in offspring. We could not find the comparable longitudinal research on this topic. Our study remains to be confirmed by future studies.

Conclusion section needs to be further improved.

Response:

We have made further edits in the discussion and conclusion, including addition of several limitations in response to the reviewers’ comments.

Round 2

Reviewer 2 Report

Authors have addressed all my comments satisfactorily and the paper can be considered for publication.